# Spatialised flood resilience measurement in rapidly urbanized coastal areas with complex semi-arid environment in Northern Morocco

Narjiss Satour [1], Otmane Raji[2,] Nabil El mocayd[3], Ilias Kacimi[1], Nadia Kassou[1]

[1] Geosciences, Water and Environment Laboratory, Mohammed V University, Rabat, Morocco.
[2]Geology & Sustainable Mining, University Mohammed 6 Polytechnic, Benguerir, Morocco.
[3] International Water Research Institute, University Mohammed 6 Polytechnic, Benguerir-Rabat, Morocco.

*Correspondence to*: Narjiss SATOUR (narjiss.satour@gmail.com)

1. **Abstract.** Enhancing resilience is critical for coastal urban systems to cope with and minimize flood disaster risks. The global increase in the frequency of floods is a significant concern for many areas in Africa. With this regard, urban planners need accurate approaches to set up a standard for measuring the resilience to floods. In Morocco, this issue is still not fully covered by the scientific community, despite the obvious need for a new approach adapted to local conditions. This study applied a composite index and geographic information system approach to measure and map resilience to floods in three northern coastal municipalities. The approach is also based on a linear ranking of resilience parameters, offering a more optimal classification of spatial resilience variation. The findings allowed to identify specific areas with different resilience levels and revealed the relationship between urban dimensions and the flood resilience degree. This approach provides an efficient decision support tool to facilitate flood risk management, especially in terms of prioritization of protective actions.

Keywords: Resilience, floods, composite index, Africa, Morocco.

## 2. Introduction

Climate change is a major challenge for the development of African countries. Several studies highlighted the severe impact of global change in Africa (Bates et al., 2008). The pattern of precipitation (Born et al., 2008; Giorgi & Lionello, 2008; Paeth, 2011), temperature (Fisher, 2015) and evapotranspiration (Speth et al.,2010) are more likely to change, which will alter the hydrological cycle, in many regions causing a change in the occurrence of extreme events such as drought and flooding (Karanja et al., 2016), especially in arid and semi-arid areas.

In particular, coastal zones situated in semi-arid are considered among the most threatened areas by the increase of flood occurrence and rapid urbanization (Leal Filho et al., 2018). Population concentration impacts flooding. Consequently, as population growth increases, exposure to floods will be a real societal problem (Kundzewicz et al., 2014). Actually, 9642 persons died out of 19,939,000 affected by floods in Africa between1993-2002 (Conway, 2009). Moreover, it is excepted that

coastal African cities will experience a higher rate of population growth and urbanization over the 21st century (UN–Habitat,2008; Lutz & Samir, 2010; Neumann et al., 2015). The rapid coastal development will exacerbate the already high vulnerability of many African coastal areas (Hinkel et al., 2011) since coastal cities are the most developed urban areas in Africa with residential, industrial, commercial, educational and military opportunities (UN–Habitat, 2015).

Morocco, situated in the North West of Africa, reveals a trend towards a decrease in average annual rainfall, as well as an
increase in average annual temperature (Hoffman et Vogel, 2008; Terink et al., 2013; El Moçayd et al., 2020). The intensity of floods will increase over time ( Vicuña et al., 2011; Doocy et al., 2013; Roy et al., 2018) while the main economic activities are located in coastal zones, where 60 % of the total Moroccan population are living (Snoussi et al., 2009). During recent years, several new policies have been implemented (Barthel and Planel, 2010; Ducruet et al., 2011; Kanai and Kutz, 2011), to improve the economic growth of these areas and reduce the negative effect of local migration. In this regard, the main drivers of the
economy are based on tourism and free zones industries, which will increase the vulnerability of these zones to climate change (Perelli, 2018). Adaptation to climate change is an important factor to consider in order to achieve sustainability in such areas. As, the combinations of environmental change, demographic growth, and urban complexity challenges will put the urban environment under pressure (Marana et al., 2019). There are several ways to tackle adaptation issues limiting the impact of a climate-related disaster and especially flooding, which is considered as the most challenging disaster (UNDRR, 2019). The
classical proposed methods to deal with such a problematic resides in implementing structural systems (Plate, 2002; Papadopoulos et al., 2017; Bertilsson et al., 2019). Sizing these systems remain subject to ubiquitous uncertainty. Climate variability will affect the reliability of such complex coastal areas systems. Therefore, adaptation should also focus on resilience (Sustainable Development Goals) (Chen and Leandro, 2019; Miguez and Verol, 2016), rather than only structural measures. Resilience approaches aim to understand and manage the capacity of a system to adapt, cope with, and shape
uncertainty (Adger et al., 2005; Folke et al., 2002).

Since the work of Holling.1973, where resilience concept originates from the field of ecology, the concept has gained increasing interest and recognition (Cretney, 2014; Weichselgartner and Kelman, 2014; Patel et al., 2017; Kontokosta and Malik, 2018). Resilience concept has been considered, in different ways, by various research fields: psychology (Westphal and Bonanno, 2007), geography (Pike, 2010; Cutter, 2010), archaeology (Redman, 2005), and physics (Cohen et al., 2000).
Recently including natural disasters, risk management, and climate change adaptation (Godschalk, 2003; Cutter et al., 2008; Gaillard, 2010; Nelson Adger& Brown, 2007; Serre et al., 2018, among others.

Within the context of disasters, and climate change, many definitions of the resilience concept have emerged. Some (Pelling, 2003; Pendall et al.,2007, IPCC, 2007;) are focusing on the ability of system, community, or city  to absorb disturbances, retaining the same basic structures and normal ways of functioning, with self-organization capacity, and adaption to stress and
change. The bouncing back to the original state (equilibrium) after a disaster is undesirable (Klein et al. 2003), social systems are in a continuous state of change. Adaptation to some new reality (Paton &Johnston, 2006) or a several states of equilibrium (Walker et al., 2004; Pendall et al., 2007), becomes one of the main characteristics of resilience depending on being able to adapt to unprecedented and unexpected changes (Ahern.2011). This is determined by the capacity of the system to organize

itself, to learn from past disasters, in the Asian Cities Climate Change Resilience Network (ACCCRN) program and to improve
risk reduction measures (UNISDR, 2015). Some previous work (Meerow et al., 2016,) linked the concept to the temporal and spatial scales, considering resilience as the ability of urban system components (ecological and socio-technical) to maintain or rapidly return to desired functions, adapting to change in the face of disturbance and quickly transform systems that limit current or future adaptive capacity. Resilience has a systemic property (Reghezza, 2015) and implies greater consideration of the time variable. Furthermore, some works (Chen N, Graham P. 2011; Colding J., & Barthel S, 2013) describe the resilience
of the system as the ability of short-term absorbing, self-organizing and long-term learning and adaptation. The abundance of definitions shared makes it difficult to have a common definition. Therefore, it is important to set a resilience definition to form a basis (Carpenter et al. 2001).

In this work, resilience of the urban system to floods is the capacity of urban-flooded areas to maintain the activities during and after floods, where a coastal urban area will be able to absorb the disaster (at an acceptable level) and adapt to the changes.
Besides, urban resilience is a complex and a multidimensional concept (Sharifi, 2016). The resilience of the urban system to floods includes several dimensions of an urban system. Social, economic, physical, natural, and institutional dimensions equally important (Batica, 2015; Qasim et al., 2016). The social dimension explores flexibility, health status, knowledge, while the economic dimension is related to the economic capacities, income resources and connections devices within the community. The physical dimension may include urban density, building materials and infrastructure (Qasim et al., 2016) or
quantified based on physical indicators such as flood depth or flood duration extracted from flood simulation data (Mugume et al.,2015; Chen and Leandro, 2019 ). Areas located at low elevation or near to the rivers are more sensitive to climatic disasters, which constitutes the natural component of resilience (Hung et al., 2016). Finally, institutions efforts aiming at coping with disasters through better planning, awareness programs and mitigation measures should be considered as the Institutional dimension (Changdeok et al., 2019).
Integrating these dimensions in the evaluation of resilience helps to have a general picture, which will lead to creating suitable management tools that can be very useful in the decision-making process (Bertilsson et al., 2019). Supporting the decision on strategies, actions and measures to be taken, planning for the long-, medium- and short-terms and assessing the progress, start with the assessment of the current and expected future status of resilience, to know where urban cities are, and helping to identify strengths and weaknesses (Cardoso et al.,2020).
Because of the multidimensional aspect, it remains challenging to quantify resilience (Bertilsson et al., 2019). Many works have shown the need to have some metrics allowing to have some measure of resilience. Yet there is no consensus about a single metric of evaluation. The literature (Meerow et al., 2016; Asadzadeh et al., 2017; Rus et al., 2018) refers to the need for measures. Making resilience tangible and practical for cities, through a transition from theory to practice is challenging (Kontokosta & Malik, 2018; Meerow et al., 2016). Quantitative approaches are used through composites indicators providing
a synthetic measurement of a complex, multidimensional, and meaningful phenomena. Those indicators are schemed based on the aggregation of multiple individual indicators (OECD, 2008). The choice of method to construct composite indices dependent upon the type of problem, the nature of the data and the goals (Nardo et al., 2005). Several composites indicators

assess urban resilience and to compare their levels within a particular geographical area (Sharifi et al., 2016; Asadzadeh et al., 2017). For example, the work of Cutter et al. (2014) using BRIC (Baseline Resilience Indicators for Communities) as the first attempt to the operationalized version of the conceptual framework "DROP model (Cutter et al., 2008). Within a socio-ecological approach, BRIC was calculated for multi-hazard context. Among other analysts, (Joerin et al., 2014 ) states CDRI (Climate Disaster Resilience Index) gauges the different capabilities needed for communities in an urban system to regain an equilibrium state after climate-related disasters such as cyclones, droughts, floods, and heatwaves. Following the same holistic spirit, the index was adopted in Climatic Hazard Resilience Indicators for Localities (CHRIL) (Hung et al. 2016). (Mayunga, 2007) also proposed a Community Disaster Resilience Index (CDRi). All of those previous indicators were applied to quantify community resilience to multi natural hazards. (Qassim et al., 2016) determined community resilience to a particular hazard "floods", and community resilience in urban areas is similar to urban resilience (Cariolet et al., 2019). Although, many particular indicators were developed for a specific case of urban resilience to a specific hazard like floods.

Based on time-dependent characteristic, (Miguez and Verol. 2016) FResI constructed to assess future resilience responses relative to the present situation. Further, (Chen and Leandro. 2019) quantified the flood resilience of households in urban areas by FRI (Flood Resilience Index) as a time-dependent method. More examples of specific indicators are available: Kotzee and Reyers (2016) spatially explicit using Geographic Information Systems (GIS) and stressing the need to move towards measuring resilience.

Regardless several challenges associated to data quality and availability constraint (Moghadas et al., 2019; Cai et al., 2018), and standard procedure for composite indicator development (Asadzadeh et al., 2017), considerable attention has been given to composite indicator (Heinzlef et al., 2019a), regarding its ability to analyze the urban, social and technical resilience of a city. However, a lack of resilience measurement tools developed by local authorities and organizations in the developing countries is revealed in a critical review (Sharifi, A., & Yamagata, Y. (2016).

Although, the Mediterranean region is a major climate change hot spot for the coming decades (Tuel and Eltahir. 2020) and Morocco is figuring out as a hotspot for climate change in several works (Born et al. ,2008; Driouech et al. ,2009; Ouhamdouch & Bahir, 2017). Moreover, the seasonal distribution of the precipitation influence strongly the Mediterranean river hydrology (Thornes et al., 2009). Assessing the intensity of the impact, Regional Climate Models (RCM) simulations over this area, all agree that Morocco might experience an increase of temperature and a decrease in precipitation (Driouech et al., 2010). Which will have a severe impact on water (Bahir et al., 2020), and natural hazards (Satta et al., 2016), among others. Consequently, increasing resilience against flooding is, therefore, of utmost importance to achieve sustainability (Snoussi et al., 2008). However, a knowledge gap for a better understanding of resilience has been identified at national and local levels (Price, R.A. 2017) in Morocco. It is highly recommended to provide policymakers with a simple approach and ways to enhance resilience to floods in the local area (OCDE 2016).

The present study is the first attempt to provide a methodological way to measure flood resilience for Northern coastal municipalities in Morocco: Martil, M'diq and Fnideq. In this work, flood resilience refers to the resilience of coastal urban areas (Martil, M'diq and Fnideq) to floods, likewise urban resilience to floods. In light of the fact that 18 hot spots located

there are highly exposed to floods (ABHL 2016). Moreover, the area is particularly highly vulnerable to multi-hazards types: floods (Karrouchi et al., 2016; Taouri et al., 2017), sea-level rise (Niazi, 2007; Snoussi et al., 2010) and coastal erosion (Satta et al., 2016; Nachite, 2009). However, the littoral is nowadays very urbanized, and tourist activities are the main economic resources in the area (Anfuso et al., 2010).

## 2. Methods: study area index development

### 2.1. Fnideq, M'diq et Martil municipalities

Related to M'diq-Fnideq prefecture, Fnideq, M'diq and Martil municipalities have a population of "984hab/km²" (RGPH 2014). Precipitation regime characterized by seasonality, annual average rainfall of 679 mm (ABHL 2016). Rainfall variability is based on altitude and the geographic situation (Karrouchi et al., 2016). Rivers flowing into the Mediterranean Sea (Martil, Mellah, Smir, Negro and Fnideq) drain slowly during the rainy months and highly in a short time during flash floods (Niazi, 2007). While, the frequency of flood events and related damages increased gradually over time (e.g. on 26 December 2000, Martil Floods have invaded more than 2400 ha in the Martil plain) (Fig.1). Urbanization is concentrated in coastal zones and puts pressure on coastal ecosystems with high touristic value (Snoussi et al.,2010). It is pitiable that municipalities are also vulnerable to multiple climate and non-climate hazards such as erosion and morphological changes (Satta et al., 2016).

### 2.2. Composite indicator development

To produce an aggregate measure of resilience, through manipulation of individual variables, constructing a "Composite indicator" is often applied. It is a mathematical combination of thematic sets of variables that represent different dimensions of a concept that cannot be fully captured by any individual indicator alone (Nardo et al., 2008).

An indicator is a quantitative or qualitative measure derived from observed facts revealing the relative position of the phenomena being measured. "It can illustrate the magnitude of change (a little or a lot) as well as the direction of change over time (up or down; increasing or decreasing)" (Cutter et al., 2010).

Moreover, considerable attention is increasingly given to composite indicators as useful tools for decision-making and public communication. To simplify and communicate the reality of a complex situation (Freudenberg. 2003) and convey information that may be utilized as performance measures (Saisana et al.,2005).

For measuring flood resilience level, contracting composite indicators has been applied (Qasim et al ., 2016; Kotzee et Reyers. 2016 ). Although, through different geographical contexts and scales, measuring resilience is significant and encompassing many theoretical perspectives.

Through exploring and analyzing the relevant literature, the quality of the framework, the data and the methodology used influence on the qualities of a composite indicator and the soundness of the messages that convey. There is a need to explain the set of steps taken to develop the Flood Resilience Index (FRI).

## 3. The theoretical comprehensiveness for primary indicator building

Flood Resilience Index is explored and calculated differently in several works: (Kotzee and Reyers 2016) used PCA as a method to construct this index and define the weights. The Flood Resilience Index construction is different for Batica. 2015 setting into account different spatial scales and focusing on urban functions. Using a time series indicators (event phase and recovery phase) (Chen and Leandro.2019) computed FRI at time t as the product of the recovery factor and the FRI at the previous time step t-1. Limiting resilience definition to two phases event and recovery (Leandro et al., 2020) developed FRI for assessing climate change adaptation.

Despite the already existing studies on flood resilience assessment, there is a lack of methods developed for a specific case of study, where data availability remains a challenge and the need of a tangible and simple way for better understanding resilience is increasing. We adopted the specific Flood Resilience Index to quantify the resilience of coastal urban areas (Martil, M'diq and Fnideq) to floods. FRI was devided into four sub-indicators: Social, Physical, Economic and Natural sub-indexes. Then, whether a sub-indicator will be included or not in the overall composite index will be identified (Fig.2). Three indicators were chosen for each sub-index (Tab1) based on data availability and its contribution in persistence, recovery or adaptative capacity (the main components of the adopted resilience definition) : Households Density (HD), Illiteracy Rate (IR) and Vulnerable Individuals Indicator (VII) were taken into consideration as the mean indicators that affect the social resilience negatively, and construct the social sub-index. The physical sub-index included the Old Buildings Rate (OBR), the Modernly Built Houses (MBH), and the Connection to Water Infrastructure (CWI). This sub-index is important because it improves the physical capacity of individual and common properties against floods, and thus minimizes their vulnerability degree. The Economic resilience sub-index also includes three indicators: Unemployment Rate (UR), Building Density (BD), and Communication Capacity (CC). Finally, Elevation (E), Stream Network Density (SND) and Distance from Depressions (DD) are the indicators selected to determine the natural resilience sub-index.

### 3.1. Selecting variables: Scoring and classification

Based on their relevance, analytical, representativeness, and accessibility, 16 variables are selected (Tab.1). The data used was mainly drawn from the National Population and Housing Census (RGPH, 2014). The Arc Hydro and Line Density modules of ArcGIS© were used to generate a stream network density from an ASTER digital elevation model (30 meters of spatial resolution), while Google high-resolution satellite imagery was used for digitizing the building area. This was converted firstly

into points, and then their density was calculated using the ArcGIS© Point Density module. The quality of data is an crucial factor that leads to realistic results (Fig.2.)

## 3.2. Normalisation

Indicators integrations into sub-indicators necessitate data transformation using data normalization. Respecting the theoretical framework and the data properties, a suitable normalisation was required; Min-Max Normalized were applied. In order to moralize the selected variable into one sub-index, each variable was normalized from 0 to 100 according to the following equations (1) and (2):

$$V^+ = \left( \frac{\text{real value} - \text{minimum value}}{\text{maximum value} - \text{minimum value}} \right) * 100 \tag{1}$$

$$V^- = \left( 1 - \left( \frac{\text{real value} - \text{minimum value}}{\text{maximum value} - \text{minimum value}} \right) \right) * 100 \tag{2}$$

The equation (1) was applied for variables that positively influence resilience while the Eq. (2) was applied to those that are negatively correlated with resilience. When the scores are attributed, each of these indicators was gridded and then a geodatabase was created in order to calculate the sub-indexes by using the GIS. Each sub-index is the mean value of all correspondent indicators.

## 3.3. Weighting and aggregation

The existing methods for determining weights are not always reflecting the priorities of decision-makers (Esty et al., 2005), that are subjective (Cutter et al., 2010). Equal-weighting is the most common for composite indices with several sub-indicators (OECD, 2008). Thus, several arguments listed by Greco et al., 2019 ("i" simplicity of construction, "ii" a lack of theoretical structure to justify a differential weighting scheme, "iii" no agreement between decision-makers, "iv" inadequate statistical and/or empirical knowledge, and, finally "v" alleged objectivity). Moreover, the weighting method selection depends on the local factors where the method is applied (Mayunga, 2007;Reisi et al., 2014). Allocating equal importance across different indicators is better suited when no knowledge exists about the interactions among the sub-indicators and composite indicator at the local scale (Cutter et al.2014; Asadzadehet al.2017). All variables are given equal weight (EW) in our case of study. The main reason is to allocate equal importance across indicators. Because of the lack of knowledge, and justification about the existing interactions among the sub-indicators and composite indicator at the local level, avoiding a large concentration of few indicators and making it is easy to communicate.

The simple method of aggregation is supposed to be transparent and easy to understand, a critical criterion for potential users (Cutter et al.,2010). All individual indicators have the same measurement unit. Therefore, using linear aggregations is preferred than geometric aggregation. The linear aggregation formula of the FRI takes the following form Eq. (3).

$$FRI = \frac{SRI+PRI+ERI+NRI}{4} \qquad (3)$$

Social Resilience Index (SRI); Physical Resilience Index (PRI); Economical Resilience Index (ERI); Natural Resilience Index (NRI); Zero is considered as low resilience level, 100 as high resilience level and 50 medium resilience level.

### 3.4. Links to other indicators

To correlate the composite indicator with related variables, statistical data analysis was performed, using the program SPSS 23. Data presented as a mean and standard deviation (st.dev) were statistically analysed using multi-variance to confront data of natural, physical, economic and social condition with the Flood Resilience Index. Furthermore, to identify which variables differ significantly between the three data sites. The significant differences were distinguished by posthoc Tukey's Honestly Significant Difference (HSD) test at $p<0.05$. The Spearman's rho coefficient was used for correlations between variables. Only correlation coefficients that were significant at a level of 0.05 are presented herein.

### 3.5. Visualization and validation

Proper attention has been given to the visualization. It helps and enhances interpretability, thought to present information graphically. Graphics and maps facilitate further exploration of geographic trends in the data (Kotzee and Reyers. 2016). Hence, to visualize FRI and sub-indicators, results were expressed using Geographic Information Systems (GIS). After visualizing the composite indicator results, validation was the last step. Acting like a 'quality assurance', robustness step will highly reduce the possibilities to convey a misleading message (Saisana et al. 2005). However, the step is often missing for the vast majority of the composite indicators (OECD.2008). Relating to resilience assessment, external validation has been used to validate several indicators (CDRI 2009, BRIC 2012, CDRI 2013, and BRIC 2014) results.

The validation based on actual outcomes in the municipalities is possible here using cross-validation type. It was performed to test and compare the reliability of FRI results in use with the results of another model used to analyze risks of hydro-climatic hazards in the local zone (Satta et al., 2016). Through exploring the opposite correlation between risk and resilience (Cutter et al., 2014; Sherrieb et al.,2010). Seeking optimization considering social and economic pathways, combining flood resilience and flood risk, measures can be effective against a broader range of hazards than when considering either method alone (Disse et al., 2020).

# 4. Results

## 4.1. Sub-indices

Each sub-index was observed separately, to get additional insights about Flood resilience Index. For the social resilience (Figure3D), produced based on the three indicators of social resilience (Figure 3A, B and C). The highest values of social resilience are more related to a few urban areas than rural and less developed sectors ones. In term of mean value, the social resilience sub-index was higher in Martil (69.03±11.24) followed by Fnideq and the coastal area of M'diq showing similar values (57.11±9.26 and 57.17±11.44 respectively).

Higher physical resilience scores (Figure 4A, B, C and D) are concentrated in the urban center areas with a spatial tendency towards the coastal area. Even though pockets of lower scores exist in the central area and some less developed sectors indicating low physical resilience levels. Therefore, the central area had a bit low level of physical resilience as compared to Fnideq, M'diq and Martil urban centres and the coastal zone (Fig.4 D).

Results (Fig.5D) show a concentration of the low and moderate level of economic resilience in the three urban centers. However, this does not exclude that some coastal urban sectors showed high levels of Economic resilience sub-index.

The overall map of Natural Resilience Index shows a spatial variability between the lowest and the medium level of NRI in the whole study area (Fig.6 D). However, the high level of natural resilience is more prevalent in areas with high altitudes, such as Capo-Negro (Fig.6 AC).

## 4.2. Total Flood Resilience Index

The results reveal a marked spatial variability of resilience to floods (Fig.7). Overall, 31% of the study area varies from low to very low, which equals 45 km² (Fig.8a). 43% of the studied area, which equivalent to 52 km2, was classified as moderately resilient and only 17% of the studied area (17 km2) was classified as highly resilient and the remaining 3% with very high resilience. The central area shows the lowest levels of FRI, including sensitive coastal sites such as Smir Lagoon, Kabila beach, and Restinga beach. In contrast, M'diq and the North of Martil have relatively moderate to high values in terms of resilience to floods. However, the significant disparities between rural and urban areas, especially in terms of socioeconomics, highly influences the flood resilience index values.

In order to avoid any confusion related to flood management priorities between the rural and the urban areas. The resilience map corresponding to urban areas were extracted, and the index values using GIS were reclassified to have the priority areas without taking into account the rural part. Using this tool to overlay the spatial distribution of households (RGPH 2014) and FRI map, it turns out that 1151 households (around 2.4%) are in areas of very low resilience and more than 7800 households (about 16%) in low-resilience areas. On the other hand, 7402 households are in a high resilience situation, and only 177 can be qualified as very high resilient (Fig.8b).

## 4.3. Statistical analysis

In order to evaluate the contribution of the sub-dimensions (Social, Economic, Physical and natural dimensions) for the resilience analysis, the statistical relationship between the total Flood Resilience Index (FRI) and its sub-indices was estimated for each municipality (Tab.2).

The SRI is positively correlated to the FRI index in the three municipalities ($p<0.001$), particularly in the urban areas where it is proven to be important as an FRI component. Regarding the ERI sub-index, it shows a moderate correlation at the Fnideq and Martil municipalities ($p<0.01$), or even a low correlation at the M'diq level ($p<0.05$). Unlike SRI and ERI, the correlation to the PRI sub-index is different from one municipality to another. It is strong at the level of Martil ($p<0.001$), weak at the level of Fnideq ($p<0.01$) and absent at the level of M'diq. In the case of the NRI sub-index, it displays a strong correlation at

the level of Fnideq and moderate at the level of Martil and M'diq.

## 5. Discussion

Within the current context of global climate change associated with an increase of flood damage, the efficient use of available data is, in most cases, the primary source of judgment control decision-making for flood risk management (Ouma et al., 2014). Producing flood resilience maps has thus become a crucial issue for local flood management planners (Godschalk, 2003).

However, these products require generally detailed knowledge about all resilience components in time and space to be effective. They should be designed in such a way that can help the decision-making by using ranking and prioritization process (Chitsaz et al., 2015). Accordingly, the choice of a good methodology to assess and quantify the resilience attains its utmost importance and relevance. Indeed, the adopted methodological approach as well as the quality of the data, has a significant influence on the obtained results, and hence on the final decision making (Suárez et al., 2016).

In this paper, the adopted methodology is adaptable according to the study case and the available data. Moreover, the adapted ranking process is based in a linear scoring, which offers the advantage to be more sensitive to changes compared to the usual methods based on assigning scores according to intervals (e.g. Angeon et al., 2015). It also provides a more reliable and objective spatial comparison of resilience parameters values which will finally allow obtaining effective prioritization of resilient areas.

It should be noted that significant components for the resilience analysis have been considered and the obtained resilience map allowed to classify the study area according to four resilience degrees to floods: very low, low, moderate and high.

The difference on the social resilience sub-index between urban and rural areas could be explained by the fact that human development indicators are generally lower in rural and less developed areas, especially those related to school attendance and the people vulnerability, which affect the social resilience negatively. However, the difference in SRI between municipalities

may occur because of the great growth rate of Martil municipality rather than Fnideq and M'diq (HCP, 2018)

The low physical resilience in the central area and the less developed sectors may exist because of the low population and urbanization (e.g. At the central area access to water infrastructure, as basic service is still low (Figure 4C). Unlike in the case of the urban centers with high physical resilience scores.

Meanwhile, the high level of Economic resilience sub-index in some coastal urban sectors may be explained by the tourist and economic activities. An expected thing as the characteristics of the wealthy residents living there (Tempelhoff et al., 2009; Kotzee et Reyers. 2016). Unlike in the three urban centers having low and moderate economic resilience. That could be explained by the high unemployment rate "17.9 %''(HCP, 2018) and the high urban density. These findings support our hypotheses and the suggestions from Cutter et al. (2010) and H.-C. Hung et al. (2016). Further, the results of (Irajifar et al. 2016) show that the association of high population density and the high incomes make a recovery after disaster quicker.

The overall picture of the natural resilience shows that all three municipalities have lower natural resilience. Martil had a bit low level of the NRI as compared to Fnideq and M'diq. This is because of the lowest values of elevation indicator and distance from depressions. The findings are fully corresponded to the existing literature (H.-C. Hung et al.,2016), supporting the relationship between elevation, flood-prone areas and the least resilience.

The areas with very low and low Flood Resilience Index seem to be generally associated with the areas showing unstable socials conditions. This observation is confirmed by the statistical analysis, and studies (Godschalk, 2003; Cutter et al., 2010; Kotzee et Reyers, 2016; Moghadas et al., 2019) showing that the social resilience is strongly correlated to flood resilience degree. Moreover, the disparities highlighted between rural and urban areas revealed that rural areas display the lowest resilience to floods.

Economic and natural resilience which is tightly linked in the sites, is the second most statistically significant indicators linked to the total FRI. Disparities between municipalities are less significant. Means that areas having low or moderate resilience to floods need equal attention (Qasim et al., 2016).

The risk and vulnerability-oriented studies (Niazi, 2007; Snoussi et al., 2010; Nejjari, 2014; Satta et al., 2016) in the coastal area were used for validation. The results are consistent, showing that coastal sites such Restinga plain, kabila beach, Smir lagoon and Martil-Alila plain having a low resilience are highly vulnerable to the flash floods and sea-level rise impacts (Snoussi et al., 2010; Niazi, 2007; Satta et al., 2016). Considering all the output, this confirms that the flood resilience index is relatively valid and can be adapted and tested in other geographical areas. Moreover, this robustness analysis makes the FRI in this case of study support the idea that areas with higher vulnerability levels examined have lower resilience levels (H.-C. Hung et al., 2016).

There is a room for improvement within the three sites. There is a need to prioritize the actions contributing to enhancing the social and economic communities' levels. Providing support and strengthen actions promoting social and economic level in the municipalities.

Further, the statistical analysis shows a significant link between the natural characteristics and resilience degrees. In that situation, it is recommended to establish best practices and measures to avoid urban development in flooded areas and to

provide more efforts to manage the risk of floods in urbanized areas, with a strong focus into the contingency plans in case of
power or drinking water failure in the three municipalities.

Therefore, there is a need to incorporate disaster management education in college to explain hazards adaptation. Also, educate people through communication devices, seminars and workshop involve citizens to be aware of the damages and the climate change effects.

The obtained results highlight the importance of using a multidimensional approach to assess flood resilience. Furthermore,
GIS is also highly recommended as a solution to complex situation and as a decision support tool that offers an interactive use and continuing improvement (Ouma et al., 2014; Mayunga, 2007).

## 6. Conclusion

Building and enhancing resilience to floods becomes critical, as the urban development in a coastal area in Africa is increasingly stressed. Especially for the coastal zones situated in semi-arid threatened areas. Nevertheless, in the local contexts
of Morocco, where this study is the first attempt focusing on enhancing the understanding of resilience to floods highlighting the application of the tangible approach to summarize and present complex components linked to resilience to floods.

Flood resilience assessment was piloted using a composite index and a GIS. The spatial and statistical analysis gave further insights into the geographic distribution of FRI across Fnideq, M'diq and Martil municipalities. Moreover, clarify the presentation of a complex set of components linked in a reproducible way.

The findings indicate that different factors can vary spatial patterns of resilience to floods. The framework is flexible enough to allow the proposed index, in future work, to take into consideration the institutional component. In order to advance our understanding of the complex nature of flood resilience, and provide useful results to suggest a floods adaptation strategies in a coastal area. The robustness of flood resilience indicator was tested by comparing the results against additional case studies and operationalized measures of resilience. However, there is no question that recommendations to improve FRI development
are suggest: starting with tackling the main limitations from considering real/simulate flood inundation maps, to integrating climatic data (flood data or flood simulation data). Besides, for robust validation, date of resilience assessment and validation tool date should be highlighted to take the specific changes in land covers between the two periods of time. Further work will use other methodologies developing FRI in the same coastal area, to provide further insights about indicators assessments and the relationships among flood resilience and flood risk.

**Acknowledgments**

The authors would like to thank the Office of the High Commission for Planning (HCP) and Hydraulic Basin Agency of Loukkos in Morocoo for making their data available for our study. Once again, we thank reviewers for the time they allowed reviewing our paper, their inputs have been precious. Special thanks are to Dr Mohammed BEN-DAOUED and Dr Mounir Ouzir and Mr Margaa Khalid for their meaningful insights provided.

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

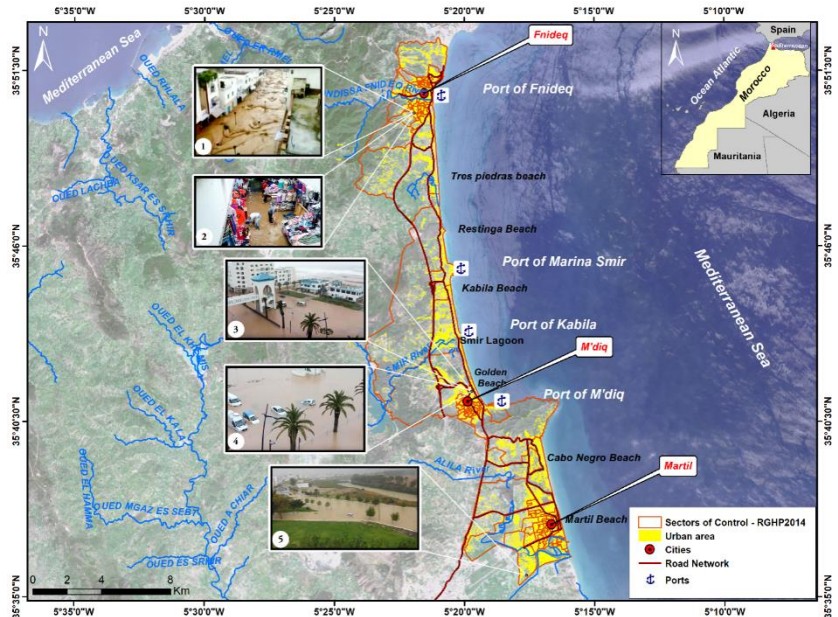


**Figure 1: Location of the three studied municipalities: Fnideq, M'diq and Martil, in Northern Morocco and examples of the flooding (1: Photo of Fnideq Center in September,28th2008; 2: Photo ofAlmassira Commercial Center Fnideqin September, 27th2014; 3 and 4: Photo of M'diq in March, 06th 2010; 5:Photo of Martil River in March, 02nd 2018). (©Copernicus data (2017).**

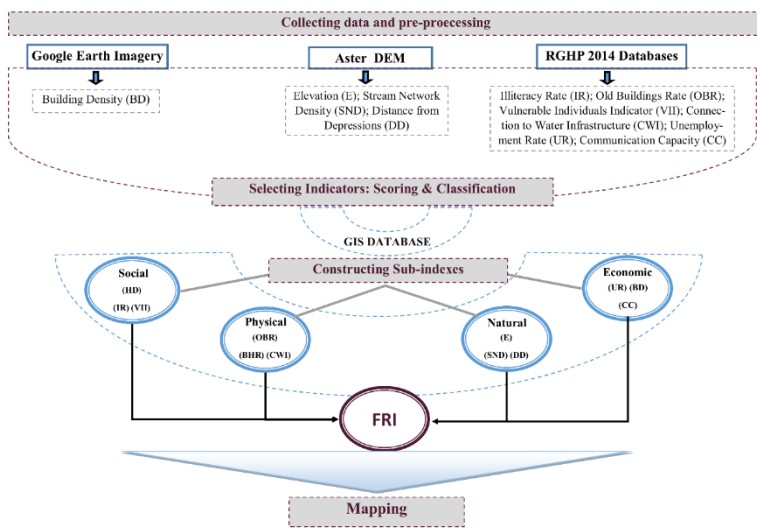

**Figure 2: Procedure used to assess flood resilience in the three municipalities**

| Dimensions | Indicators | Description, effect on resilience &justification |
|---|---|---|
| | Households Density (HD) | Cities with higher building density in developing countries tend to be densely populated, with many areas that have grown fast, |

| | | |
|---|---|---|
| Social (SD) | | (Andersson, 2006), often with insufficient infrastructure, resulting in environmental degradation and high damaging floods. Studies have found that high resilient sites had low population density (Sanabria-Fernandez et al., 2019). |
| | Illiteracy Rate (IR) | The persons who have never learned to read. That can make the emergency and public awareness processes challenging. (Cutter et al., 2010) |
| | Vulnerable Individuals Indicator (VII) | It refers to all vulnerable people (0-14 year olds, 60 year olds and disabled people) who can creates hindrances in mobility during floods and operations of evacuation ( Hung et al., 2016; Qasim et al.,2016). |
| Physical (PD) | Old Buildings Rate (OBR) | Is the percentage of buildings that are over 50 years old, it expresses the fragility that increases with building materials age. |
| | Modernly Built Houses (MBH) ) | Based on the building material factor (by Reinforced concrete and bricks with mortar) modernly built houses will suffer less exterior damage during floods events in the local state (Cutter et al., 2010). |
| | Connection to Water Infrastructure (CWI) | The rate of connection to the sewage system and drinking water distribution strength resilience community (Cutter et al., 2010). A not being guaranteed access to water during and after emergency (Pagano et al., 2017) will aggravate the situation. |
| Economic (ED) | Unemployment Rate (UR) | It expresses the decrease in the individual economic capacity. Unemployed people are faced with difficulties related to their disability to recover or rebuild their damaged property (Cutter et al., 2010; Sherrieb et al., 2010). |
| | Building Density (BD) | It reflects the concentration of building per area. People are more concentrated in low quality urban housing, infrastructure and services the impact of natural disaster is higher (Pallard et al., 2009).It was selected based in the fact that an area with high building density is less resilient to floods. |
| | Communication Capacity (CC) | Is the rate of persons having communication devices (Television, Mobile phone and Internet).It express communication facilities availability , during, after and before flood hazards. strengthen resilience (Cutter et al., 2010). |
| Natural ( ND) | Elevation (E) | It was selected based on the fact that lands with low elevation, are more risked to flooding and exposed to damages compared to high elevation areas. |

| | Stream Network Density (SND) | It describes the degree of drainage network development and was recognised to be significantly linked with the formation of flood flows (Pallard et al., 2009). |
| | Distance from Depressions (DD) | It expresses the distance from flood-prone areas or flood risk areas (ABH databases 2016) including natural depressions of high flow accumulation. |

**Table 1: Indicators descriptions selected to assess the flood resilience in Fnideq, M'diq and Martil area; (compiled from different sources)**



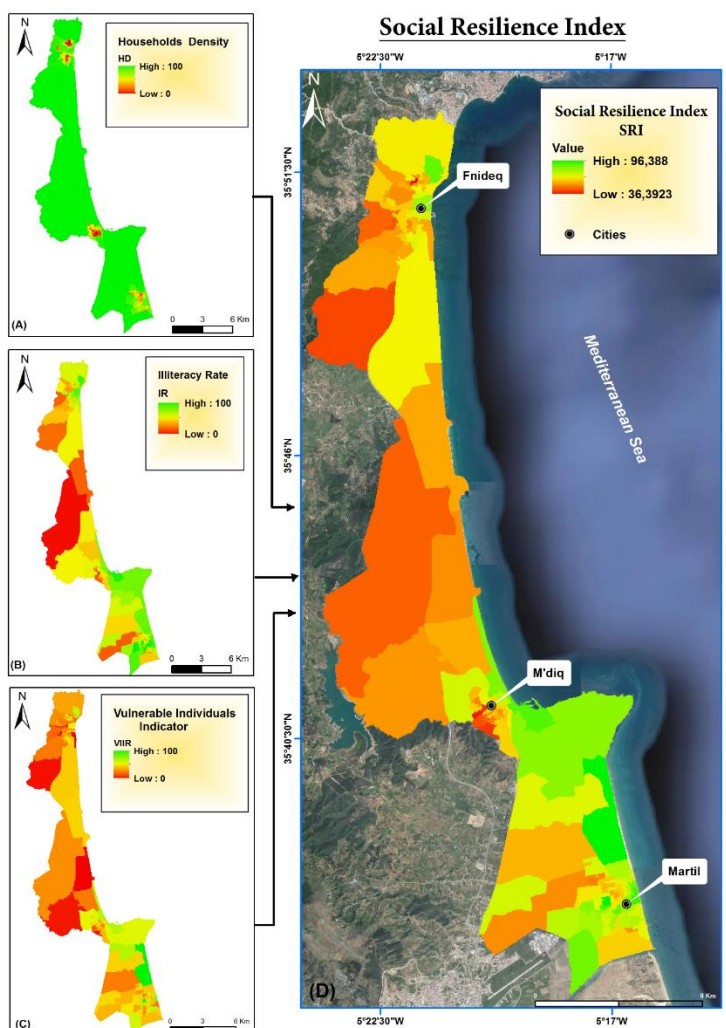

**Figure3: Spatial distribution of A: Households Density, B: Illiteracy Rate; C: Vulnerable Individuals Indicator and D: Social Resilience Index (obtained from © Google map image in 2018).**

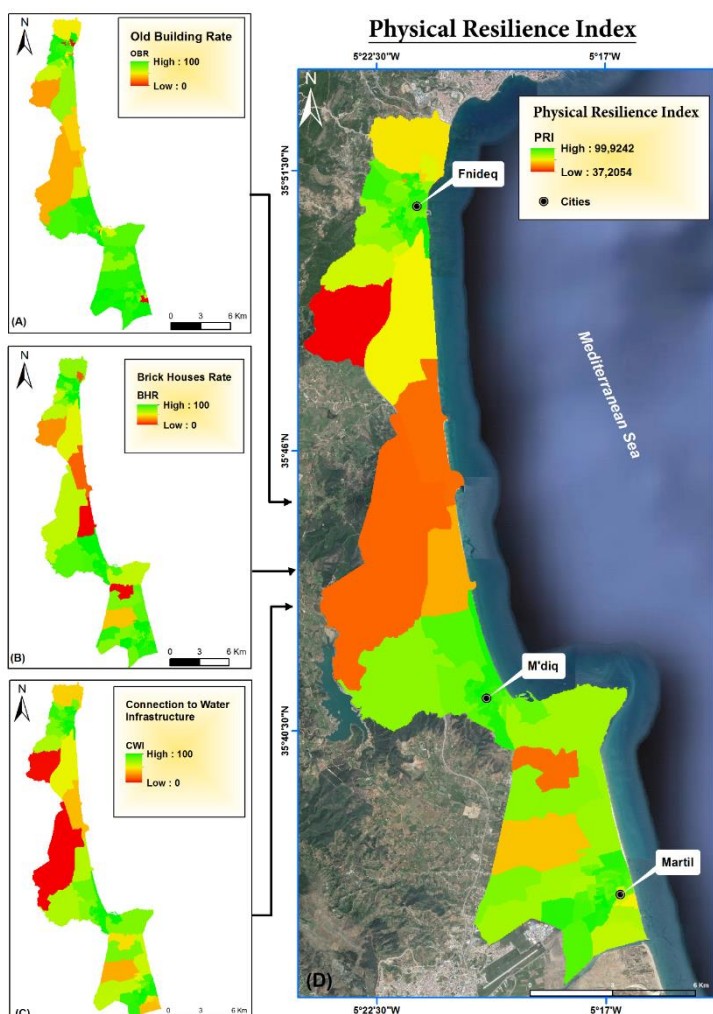

**Figure 4: Spatial distribution of A: Old Buildings Rate, B: Brick Houses Rate, C: Connection to water infrastructure and D: Physical Resilience Index (obtained from © Google map image in 2018).**

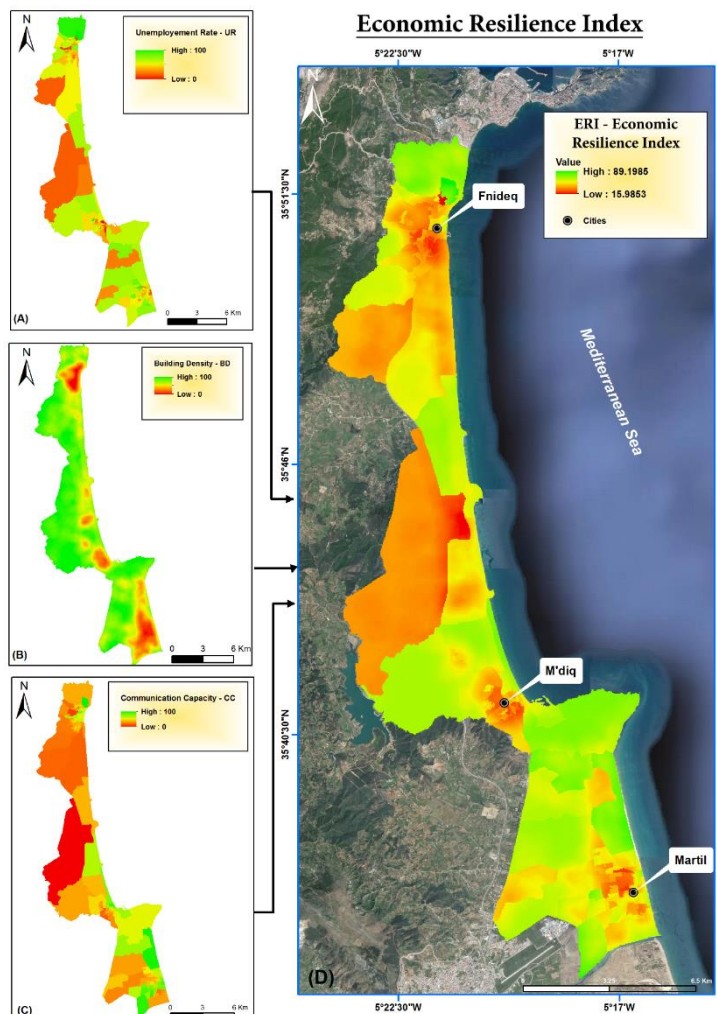

**Figure 5: Spatial distribution of A: Unemployment Rate, B: Building Density 2017, C: Communication Capacity and D: Economic Resilience Index (obtained from © Google map image in 2018)**

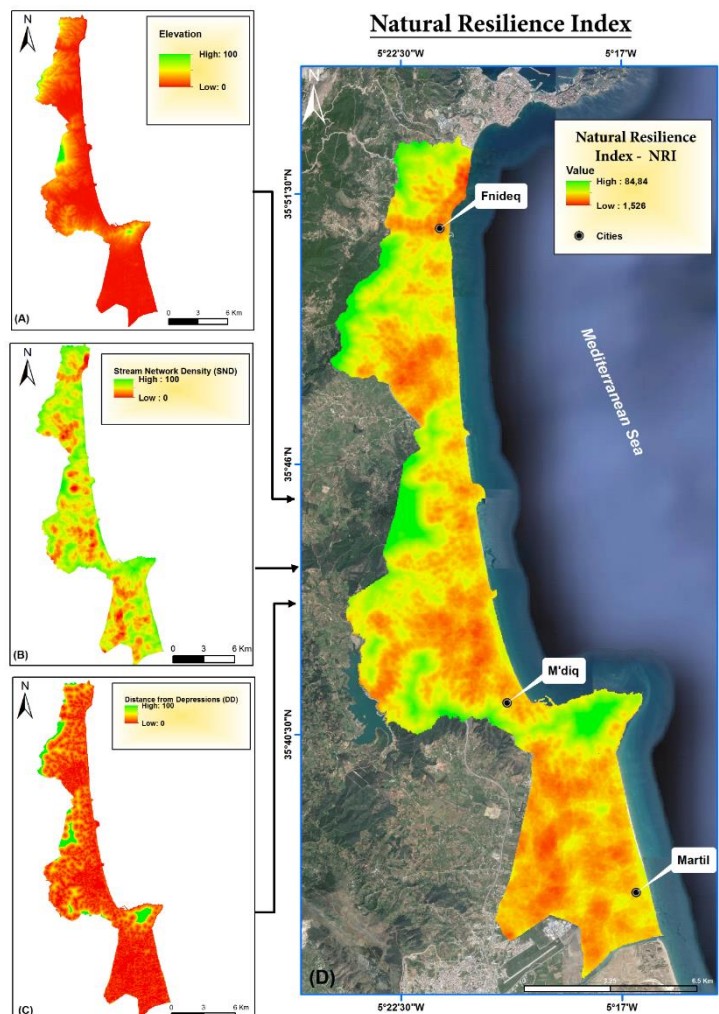

**Figure 6: Spatial distribution of A: Elevation, B: Stream Network Density, C: Distance from Depressions and D: Natural Resilience Index (obtained from © Google map image in 2018)**

|  |  | SRI | ERI | PRI | NRI |
|---|---|---|---|---|---|
| **FRI** | **Fnideq** | 0.643*** | 0.441** | 0.378* | 0.650*** |
|  | **Martil** | 0.764*** | 0.425** | 0.589*** | 0.470** |
|  | **M'diq** | 0.800*** | 0.408* | - | 0.544** |

*p<0.05; **p<0.01; ***p<0.001.

**Table 2: Spearman's rho Correlation between the total Flood Resilience Index (FRI) and its dimensions.**

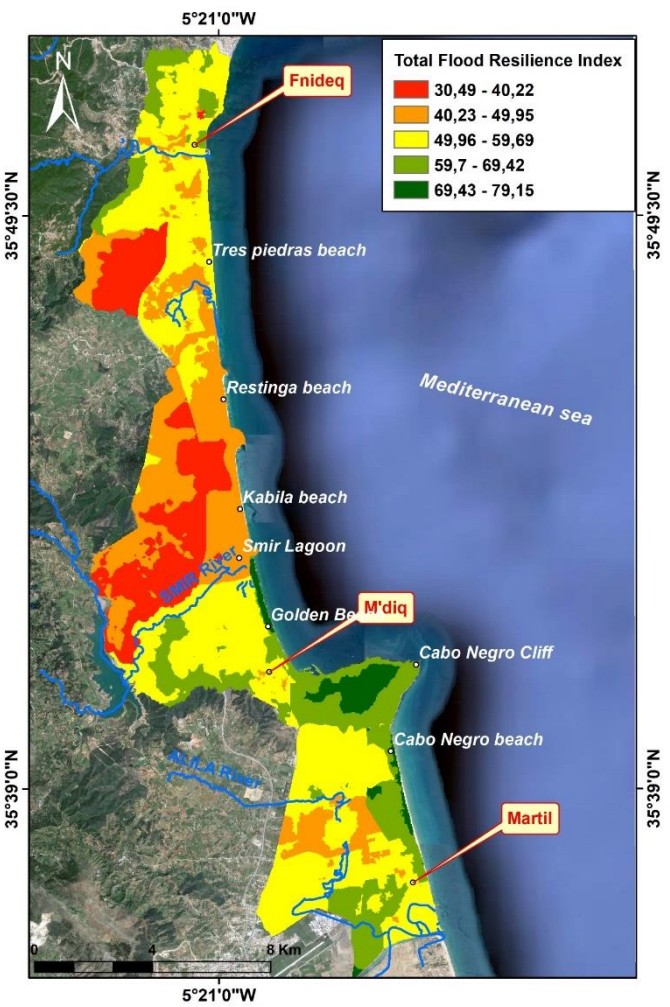

**Figure 7: Distribution of Total Flood Resilience Index. (obtained from © Google map image in 2018)**

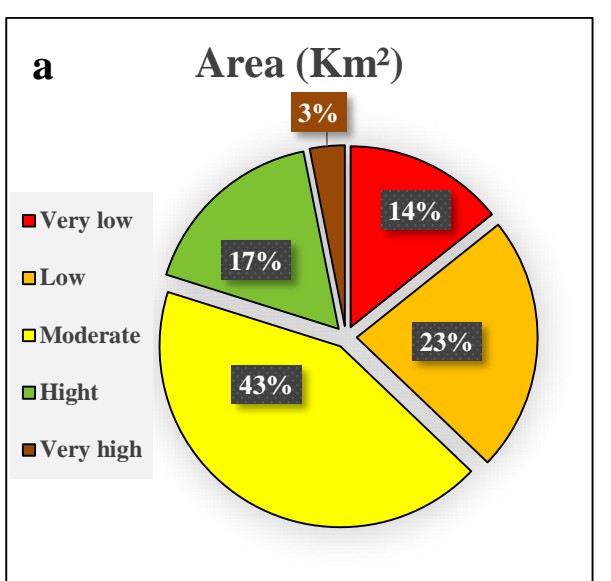
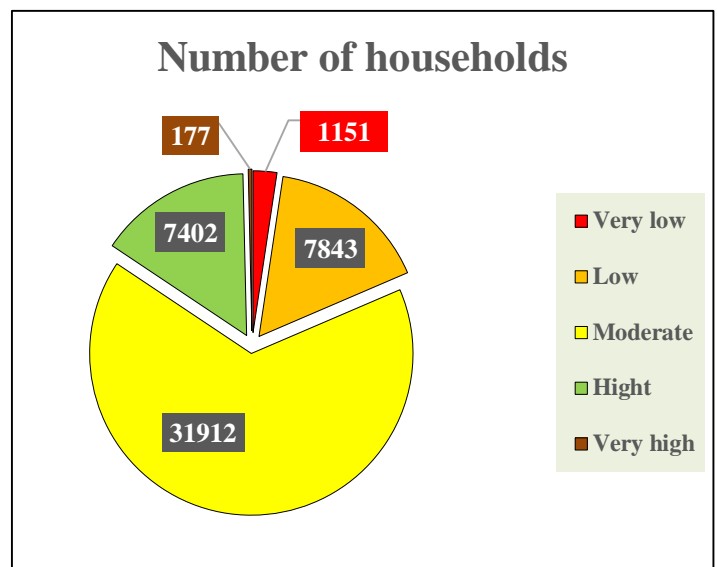

**Figure 8: a) Total Flood Resilience scores distribution according to the surface of the study area; b) Total Flood Resilience scores distribution according to Households numbers in the study area.**