# Peer review of "Spatialised flood resilience measurement in rapidly urbanized coastal areas"

_Natural Hazards and Earth System Sciences, 2019_

## Referee Comment (RC1) · Jorge Leandro (Referee) · 28 Jan 2020

The paper presents an interesting study on flood resilience for coastal areas. The paper reads very well. I have one major concern which leads to few references that must be added to the literature review on this topic.

My major concern is related with the fact that the study is solely based on geographical data excluding flood data. Event dough the focus is on flood resilience; resilience is not based on flood data or flood simulation. In my point of view, this is an important drawback that needs to be addressed in several sections of the manuscript (including abstract and conclusion).

In the introduction a paragraph needs to be added on resilience and its dimensions. Particular the physical dimension, is often quantified based on physical indicators such as flood depth or flood duration (https://doi.org/10.1016/j.watres.2015.05.030, and https://doi.org/10.3390/w11040830) extracted from flood simulation data. The advantage of the latter reference is that recovery (one important stage of resilience) is time variable and can last longer that the flooding event itself.

The second paragraph in the introduction should be complemented with studies on existing climate change adaptation focusing on resilience (https://doi.org/10.1177/0265813516655799, https://doi.org/10.1016/j.watres.2020.115502).

Also a section on the limitations of the method presented should be added. This should state the drawbacks of an approach that does not consider real/simulate flood inundation maps. For example, The index of Elevation, would not be necessary. Since the flood routes would be captured by the flood maps. Those would reveal that some low lying areas may be not so flood prone than high lying areas because they may or may not lay near to a flood route.

Also I am unsure (line 192) what is meant with dam area. Is a dam area a flood risk area? If we consider that connection to a sewer system is enhancing our resilience why is a dam area the opposite? As far as I understood, there is no failure mechanism in this work, hence both should tend in the same direction.

One particular section I liked was 3.5. It includes a sentence relating risk and resilience. Are they really opposite? Perhaps the Authors could extent that paragraph. A recent paper discussing that point has been recently published, and may be worth discussing here (https://doi.org/10.1016/j.wasec.2020.100059).

I have no further comments regarding the text or the figures, both are already of high quality.

---

## Author Comment (AC1) · 13 Feb 2020

Dear Jorge Leandro We are deeply grateful for all the relevant remarks, interaction and the time you allowed for reading our article and giving us positive and constructive feedback. Please find bellow answers on your comments. We added your suggestions references to the text and followed your remarks.

- Including flood data or flood simulation "My major concern is related with the fact that the study is solely based on geographical data excluding flood data. Event dough the focus is on flood resilience; resilience is not based on flood data or flood simulation. In my point of view, this is an important drawback that needs to be addressed in several

sections of the manuscript (including abstract and conclusion)"

Regarding the major concern of using "solely a geographical data excluding flood data", we totally agree with the fact that including a flood or meteorological data is highly important and will give more authenticity. Unfortunately, there is no available data or flood map at the area of the study. However, we used the flood risk hot spot area geolocalisation, data we got from the official sources (Hydraulic Basin Agency- ABH 2016) to calculate the distance from depression parameter (DD). Recent references figuring out on the text (Karrouchi et al., 2016 and Taouri et al., 2017) described and discussed the flood phenomenon. And more extensive works focusing on mutihazards risk (Satta et al., 2016) and sea-level rise (Snoussi et al., 2010) and (Niazi, 2007) were helpful. Nonetheless, your proposition about mentioning the drawback in sections of the manuscript will be taken into account.

- Introduction "In the introduction, a paragraph needs to be added on re-silience and its dimensions. Particular the physical dimension, is often quan-tified based on physical indicators such as flood depth or flood duration (https://doi.org/10.1016/j.watres.2015.05.030, and https://doi.org/10.3390/w11040830) extracted from flood simulation data. The advantage of the latter reference is that re-covery (one important stage of resilience) is time variable and can last longer that the flooding event itself."

For the resilience dimensions, we will add a paragraph (Line 95-107) and the suggested references (Line 101-102). The second paragraph in the introduction should be com-plemented with studies on existing climate change adaptation focusing on resilience. Other references will be added accordingly in the revised Manuscript.

- Limitations "Also a section on the limitations of the method presented should be added. This should state the drawbacks of an approach that does not consider real/simulate flood inundation maps. For example, the index of Elevation, would not be necessary. Since the flood routes would be captured by the flood maps. Those

would reveal that some low lying areas may be not so flood prone than high lying areas because they may or may not lay near to a flood route."

Drawbacks and limitations of the method and data used will be discussed in the conclusion. (Line 452-431)

"Also I am unsure (line 192) what is meant with dam area. Is a dam area a flood risk area? If we consider that connection to a sewer system is enhancing our resilience why is a dam area the opposite? As far as I understood, there is no failure mechanism in this work, hence both should tend in the same direction."

The designation "dam area" is the flood risk area as you mentioned. We used data we got from the official sources (Hydraulic Basin Agency- ABH 2016) to calculate the DD. This will be highlighted in the revised form of the Manuscript.

"One particular section I liked was 3.5. It includes a sentence relating risk and resilience. Are they really opposite? Perhaps the Authors could extent that paragraph. A recent paper discussing that point has been recently published and may be worth discussing here (https://doi.org/10.1016/j.wasec.2020.100059)."

The authors would like to explore the correlation between resilience and risk in the context of the cross-validation step. However, the "opposite" correlation depends on the spatial and temporal scale. Resilience is locational and context-specific. Otherwise, the relationship may "not be opposite" in case of another geographical area. Or the same geographical area, with more or another database or resilience assessment methodology. The relationship between risk and resilience worth discussion in the academic literature. The reference suggested and others will be added to the Manuscript. Thank you!

---

## Referee Comment (RC2) · Anonymous Referee #2 · 17 May 2020

Dear authors,

I carefully read your paper which focuses on the important subject of resilience of urbanized coastal areas. I will first give some general comments and then more specific comments and discussion points

General comments

In general, the subject is within scope of NHESS and the subject is scientifically relevant. The paper aims to provide a new method and results for an area which is not frequently studied: the coastal areas of Morroco. The paper aims to quantify resilience in order to allow monitoring and to identify more and less resilient areas to flooding

and climate change. However, the method for quantifying is not well motivated and not clearly linked to the very brief review and resilience definition provided before. It is not possible to evaluate the outcomes on their plausibility or to know if they are transferable to other areas. Also the link to measures or policy making or the relevance of the outcomes is not touched upon. Furthermore, it is not clear how this Flood Resilience Indicator complies with or differs from existing indicators. Finally, the english needs significant improvements and not all references mentioned are in the list of references. The format of the references does not comply with the NHESS standards.

It is recommended that the authors explain how they define resilience, provide an overview of existing frameworks or indicators and discuss why they use those or not and then provide and motivate their own indicators and explain what is the innovative value compared to existing indicators or frameworks. It should be clear how the indicators and outcomes relate to their resilience-view. Finally, the results and their value for the region should be discussed, their plausibility assessed and lessons or implications for other areas should be described. The paper is already a good starting point. The resilience of coastal communities to floods in the context of sea level rise is an important subject where probably many flood risk managers are interested in and struggling with.

Discussion points

1. How do the authors define resilience? Since the paper is on measuring resilience, the authors should define what is meant by resilience and what they aim to measure: resilience of what to what? This is not clear in the current paper. The authors first seem to have adopted the resilience definition of Adger et al.(2005) and Folke et al, (2002): "Resilience approaches aim to understand and manage the capacity of a system to adapt to, cope with and shape uncertainty", but then they mention that many definitions in various fields exist (which is true, but we need to know how the authors define it here). In line 94 they refer to urban resilience instead of flood resilience and the paragraph ends with a sentence on 'community disaster resilience frameworks'.

Although it is mentioned that indicators and frameworks exist, they are not provided or discussed. It is also not clear if the authors consider urban resilience as similar to disaster risk resilience and flood resilience.

The resilience view is also not clear on line 102 where resilience is called a "multidimensional objective", while in line 103 resilience is called "an approach". Do the authors see resilience as an aim/objective or an approach or a means to reach an aim (e.g. a better coastal community, smaller flood impacts or better functioning economies)? Or both? Then again references to various indicators are provided, while the indicators itself are not mentioned. The paper then mentions that there is a gap in knowledge on how to measure resilience, but also concludes that resilience needs to be enhanced, so some knowledge on the current resilience is present: at least enough to conclude that the resilience is currently insufficient.

Then from line 113 onwards it is not clear whether the paper looks at resilience to floods which may be affected by sea level rise or coastal erosion or the resilience to floods, sea level rise and coastal erosion all together. That should be clarified. On line 159 the paper states that resilience assessments can be classified into measuring persistence, recovery and adaptative capacity. This makes the concept more concrete. However, these three terms or this distinction is not referred to anywhere else in the paper. Why did the authors put this sentence there? How does it relate to the proposed Flood Resilience Index?

2. The Flood Resilience Index used in the paper: how does it relate to existing frameworks? In chapter 3 on line 164 the flood resilience index is mentioned. Is that new and is that what the authors have developed? Is it related to the indicators and frameworks to which the authors have referred to ? How, or why not?

3. The indicators and subindicators itself: The choice for the indicators is not motivated well. The authors state for example that areas with a higher building density are less resilient. Why is that? Or is that true in Morocco? Is it because the flood impacts may

be higher than in rural areas or areas with less exposure? But perhaps there are also more funds to recover from that damage? Why are areas with a better connection to sewage or drinking water system more resilient to floods (or to floods, coastal erosion and sea level rise, that is not clear in this paper)? There is a reference to Cutter there, but Cutter describes disaster resilience, and not flood resilience, which may be different. Why is communication capacity an economic indicator and not a social one? Is it fair to count both the percentage of old houses, and the percentage of modern houses or is that double counting the same aspect? Is there a storyline to explain the indicators selected: how does unemployment rate, relate to flood resilience ( I assume because less funds will be available for a quick recovery?, or is it based on statistical analysis of this factor and flood recovery? Or flood impacts?)

4. What is the use and what are the limitations of such a composite indicator: What if two areas would have the same low score, but one has a low score because it has many persons below 14 or above 60, while the other area has a low value because of it's low elevation, how would you use that score? What would be the value of a composite indicator if causes of low resilience could be completely different and therefore also solutions or measures may be very different? What is the value for an area without inhabitants? (flood-prone or not) and what would be the value for a densely populated area which is not flood-prone? And what is the value for an area where floods cause impacts which are overcome within a year, or where sea level rise scenarios for the next 50 years can be coped with without a significant increase of flood risks? These questions are related to flood resilience, aren't they? How does this indicator relate to those? Why do you value all subindicators equally?

5. The English and the writing style The English needs significant improvements. Sometimes sentences start with 'While x and y is going going on.. and then they end without a second part of the sentence. Some parts are repeated several times e.g. that resilience is often quantified by composite indicators (around line 101, 137, 145, 149)

6. References: The authors provide many references in the review, however sometimes improvements are needed. Sometimes the relation or the link between the referenced work and the work of the authors is not clear (e.g. if stated that they have a resilience indicator, it is not explained what indicator, whether it is useful or not and why, just that there is an indicator), some references are missing in the reference list (e.g. Lutz & Samir, 2010) and some are perhaps less relevant? (e.g. for the claim that floods will occur more frequently in Morroco there are 3 references, one relates to a paper on climate change impacts on hydropower systems in California and is probably less relevant than the other 2). The reference formats are not in line with the journal's requirements.

---

## Author Comment (AC2) · 13 Jun 2020

We want to thank the reviewer for the constructive comments, which will surely improve the quality of this paper. We appreciate the quality of the reviewer's questions. All the comments will be positively considered in the revised Manuscript. Please find our responses to the comments raised in the discussion point of the review:

Discussion Point

Remark 1

**How do the authors define resilience? Since the paper is on measuring resilience,

the authors should define what is meant by resilience and what they aim to measure: resilience of what to what? This is not clear in the current paper. The authors first seem to have adopted the resilience definition of Adger et al.(2005) and Folke et al, (2002): "Resilience approaches aim to understand and manage the capacity of a system to adapt to, cope with and shape uncertainty", but then they mention that many definitions in various fields exist (which is true, but we need to know how the authors define it here).

=> As mentioned by the reviewer in the general comments, this work aims to measure the resilience of the urban system to floods. Resilience quantification allows monitoring and identification of the more and less resilient areas to flooding. From our point of view the resilience concept must address the following questions: 'resilient of what?' and 'up to what level?' (Carpenter et al. 2001). The definitions given by Adger et al. (2005) and Folke et al. (2002) are general and cover our specific definition of resilience. In developing countries, the lack of statistically robust data is the ultimate challenge, especially with the upcoming climate change impact. Resilience is the ability of urban flooded areas to maintain the activities during and after floods, where a coastal urban area will be able to absorb shocks (at an acceptable level) and adapt to the changes.

Carpenter S., Walker B., Anderies J.M., and Abel N. (2001). From metaphor to measurement: Resilience of what to what? Ecosystems, 4,765-781

** In line 94 they refer to urban resilience instead of flood resilience and the paragraph ends with a sentence on 'community disaster resilience frameworks'

=> This will be corrected.

** Although it is mentioned that indicators and frameworks exist, they are not provided or discussed. It is also not clear if the authors consider urban resilience as similar to disaster risk resilience and flood resilience

=> A paragraph providing a discussion of the most used indicators will be added. In the

context of this work, yes, they are similar in our paper. Indeed, flood resilience measurement in the urbanized coastal area is the aim. Therefore, "urban resilience" refers to the coastal urban area exposed to floods. Flood resilience refers to the resilience of these urban areas to floods.

** The resilience view is also not clear on line 102 where resilience is called a "multidimensional objective", while in line 103 resilience is called "an approach". Do the authors see resilience as an aim/objective, an approach, or a means to reach an aim (e.g. a better coastal community, smaller flood impacts, or better functioning economies)? Or both?

=> "Multidimensional objective" in Line 102 and "The approach aims to provide a synthetic measurement" in line 103 both refer to the composite indicator (line 101) which is an approach aiming at providing a synthetic measurement of resilience.

** The paper then mentions that there is a gap in knowledge on how to measure resilience, but also concludes that resilience needs to be enhanced, so some knowledge on the current resilience is present: at least enough to conclude that the resilience is currently insufficient.

=> The "gap in knowledge on how to measure resilience" in our paper refers to the specific case of Morocco (and could be extended to other similar countries) where quantifying resilience needs to be adopted and enhanced based on developed countries experiences. This does not negate the existence of knowledge or research on this subject.

** Then from line 113 onwards, it is not clear whether the paper looks at resilience to floods which may be affected by sea-level rise or coastal erosion or the resilience to floods, sea-level rise, and coastal erosion together. That should be clarified.

=> Our objective deals with resilience to floods, not coastal erosion nor sea-level rise. More clarification will be added in the introduction to avoid any possible confusion.

** On line 159 the paper states that resilience assessments can be classified into measuring persistence, recovery, and adaptative capacity. This makes the concept more concrete. However, these three terms or this distinction is not referred to anywhere else in the paper. Why did the authors put this sentence there? How does it relate to the proposed Flood Resilience Index?

=> More clarification will be added to the corrected version of the paper.

Remark 2

** The Flood Resilience Index used in the paper: how does it relate to existing frameworks? In chapter 3 on line 164, the flood resilience index is mentioned. Is that new and is that what the authors have developed? Is it related to the indicators and frameworks to which the authors have referred to? How, or why not?

=> More clarifications will be added, explaining how the FRI is related to the existing frameworks in chapter 3.

Remark 3

** The indicators and sub-indicators itself: The choice for the indicators is not motivated well.

=> More details will be added in section 3 to motivate the choice of the indicators.

** The authors state for example that areas with a higher building density are less resilient. Why is that? Or is that true in Morocco? Is it because the flood impacts may be higher than in rural areas or areas with less exposure? But perhaps there are also more funds to recover from that damage?

=> In many parts of the world, higher building density, especially in developing countries (like Morocco) tend to be densely populated, with many areas that have grown fast, often with insufficient infrastructure, resulting in environmental degradation and high damaging floods. That is why in this study we consider higher building density

areas as less resilient areas. Some references will be added.

** Why are areas with a better connection to sewage or drinking water system more resilient to floods (or to floods, coastal erosion, and sea-level rise, that is not clear in this paper)? There is a reference to Cutter there, but Cutter describes disaster resilience, and not flood resilience, which may be different.

=> Water drinking access and sewage connection are human development signs in developing countries. They are reflecting a certain social resilience against all kinds of disaster effects. Naturally, they also reflect social resilience to the impacts of the floods. A not being guaranteed access to water during and after floods may imply an inequitable aggravation of the situation. For example, using non-potable water after flood disasters evolves numerous health risks. This will be more clarified in the revised manuscript.

** Why is communication capacity an economic indicator and not a social one?

=> Communication can surely be viewed as a social component. However, in this study, we consider it as an indicator of the economic situation of the population. Wealthy people in countries like Morocco have more access to communication. This population can indeed remain better informed before, during, and after flood events.

** Is it fair to count both the percentage of old houses and the percentage of modern houses or is that double-counting the same aspect?

=> The old Buildings rate (OBR) and the Modernly Built Houses (MBH) aren't representing the same aspect. The first one is based on the age factor, while the second is based on the building materials. More sentences will be added to clarify this point.

** Is there a storyline to explain the indicators selected: how does unemployment rate, relate to flood resilience (I assume because less funds will be available for a quick recovery?, or is it based on statistical analysis of this factor and flood recovery? Or flood impacts?)

=> This is true. Unemployment is related to flood resilience because less funds will be available for a quick recovery, as it's mentioned on the tab "Unemployed people are faced with difficulties related to their disability to recover or rebuild their damaged property (Cutter et al., 2010; Sherrieb et al.,2010). This will be clarified in the upcoming version of the paper.

Remark 4:

** What is the use and what are the limitations of such a composite indicator: What if two areas would have the same low score, but one has a low score because it has many persons below 14 or above 60, while the other area has a low value because of it's low elevation, how would you use that score? What would be the value of a composite indicator if causes of low resilience could be completely different and therefore solutions or measures may be very different? What is the value for an area without inhabitants? (flood-prone or not) and what would be the value for a densely populated area which is not flood-prone? And what is the value for an area where floods cause impacts which are overcome within a year, or where sea level rise scenarios for the next 50 years can be coped with without a significant increase of flood risks? These questions are related to flood resilience, aren't they? How does this indicator relate to those? Why do you value all sub-indicators equally?

=> This is an important point, which is classically discussed in the community when choosing between the equal-weighted or non-equal-weighted composite index combinations. In our case, the reasons for the equal-weighted choice have been briefly mentioned in section 3.3. here we will try to resume the discussion about this question more explicitly. First, equal-weighting is the most common for composite indices with several sub-indicators (OECD, 2008) because of several arguments listed by Greco et al.2019 ("i" simplicity of construction, "ii" a lack of theoretical structure to justify a differential weighting scheme, "iii" no agreement between decision-makers, "iv" inadequate statistical and/or empirical knowledge, and, finally "v" alleged objectivity). In addition, allocating equal importance across different indicators is better suited when no knowledge exists about the interactions among the sub-indicators and composite indicator at the local scale (Cutter et al.2014; Asadzadehet al.2017). We will add these details to the upcoming version of the Manuscript.

Regarding the question of what if two areas would have the same low score, but one has a low score because it has many persons below 14 or above 60, while the other area has a low value because of its low elevation, how would you use that score?

We believe that resilience depends on the location and on the context. Moreover, decisions made by stakeholders have also a direct impact on the resilience level. In our approach, we have taken into account these details in the design of the composite index in such a way that it is modular and adaptable accordingly. Finally, the remark about the limitations remains relevant. The limitations will be more developed on the manuscript with some discussion related to data availability and the integration of the climatic data (flood data or flood simulation data) and the validation step.

Asadzadeh, A., Kötter, T., Salehi, P., & Birkmann, J. (2017). Operationalizing a concept: The systematic review of composite indicator building for measuring community disaster resilience. International journal of disaster risk reduction, 25, 147-162. Cutter, S. L., Ash, K. D., & Emrich, C. T. (2014). The geographies of community disaster resilience. Global environmental change, 29, 65-77. Greco, S., Ishizaka, A., Tasiou, M., & Torrisi, G. (2019). On the methodological framework of composite indices: A review of the issues of weighting, aggregation, and robustness. Social Indicators Research, 141(1), 61-94. OECD. (2008). Handbook on constructing composite indicators: Methodology and user guide. Paris: OECD Publishing.

Remark 5

** The English and the writing style The English needs significant improvements. Sometimes sentences start with 'While x and y is going on.. and then they end without a second part of the sentence. Some parts are repeated several times e.g.that resilience is often quantified by composite indicators (around line 101, 137, 145,149)

=> A full proofreading of the English will be done in the revised form.

Remark 6

** The authors provide many references in the review, however sometimes improvements are needed. Sometimes the relation or the link between the referenced work and the work of the authors is not clear (e.g. if stated that they have a resilience indicator, it is not explained what indicator, whether it is useful or not and why just that there is an indicator), some references are missing in the reference list (e.g. Lutz & Samir, 2010) and some are perhaps less relevant? (e.g. for the claim that floods will occur more frequently in Morroco there are 3 references, one relates to a paper on climate change impacts on hydropower systems in California and is probably less relevant than the other 2). The reference formats are not in line with the journal's requirements.

=> Some irrelevant and related remarks will be revealed and improved. The format of the references will comply with the NHESS standards in the revised version of the paper.

---

## Referee Report (RR1)

**Title**: Spatialised flood resilience measurement in rapidly urbanized coastal areas with complex semi-arid environment in Northern Morocco

**MS No.**: nhess-2019-417

**Dear Author and co-authors,**

It was a pleasure to read and review your manuscript.

You have an interesting manuscript with some very good dataset that you have produced even the difficulty of the availability of input data. The manuscript has been written in a simple scientific language that everyone can understand easily. It definitely shows the skills of a good scientific writer.

The flood resilience of the 3 municipalities has been dealt with from different angles making the results more interesting. I have a few remarks and comments, and I hope they will be useful.

I am sure that this article will represent an important reference for several works in the Nord African regions that have the same situation and problematic.

Wish you all the best,

**Remarks and comments:**

Methodology

1.  Households Density (HD), Old Buildings Rate (OBR) and Natural (ND) are a new indicators, right? In the text, it's difficult to identify the ones that are susceptible to be taken in this study areas from literature and the new once that you considered.

2.  What are the advantages of adopting these particular case studies over others in this case?

3.  It is mentioned in …. historical records of 2014 are taken. Why are more recent data not included in the study? Are there any changes to situation in recent years?

4.  It is mentioned, "To avoid subjectivity an equal weighting was adopted" . How you gave them equal weights while each parameter might have different importance level?

Results and Discussion

1.  It is mentioned, that "the high level of natural resilience is more prevalent in areas with high altitudes". Did you get any observation about this relevant? Because it's not always true, some areas with high altitude can be affected by landslide and slumping during floods.

2. Are there a strong relationship between vulnerable area and resilience, can you say that all the areas with high vulnerability to the flood have automatically low resilience?

3. Can you globalize the conclusions of this work on the Nord Mediterranean Moroccan coast that are quite similar in different aspect, or is it just particularly adapted to those tree municipalities? If the response is yes, it will be interesting to mention that in your conclusions, if no please give the raison.

4. Your results is the combination of all the indicators that you establish for this work, in your opinion whish one represent the important factor for resilience in the Moroccan Mediterranean coast area ?

5. I know that this subject is new research subject, especially in Nord African region, have you looked to projects in the same contexts in semi-arid coast region? It will be interesting to discuss your results with their conclusions.

**Please find bellow some corrections related to spelling**

**Affiliation:** Mohammed V University in Rabat

**Line** 51: (1970)
63: (Ahern, 2011).
69 : (Chen N, & Graham P. , 2011; Colding J., & Barthel S, 2013)
72: al.,
104: (Hung et al., 2016). (Mayunga, (2007)
106: (Qassim et al. (2016)
109: (Miguez and Verol (2016)
110 : (Chen and Leandro ( 2019)
118: (Sharifi, A., & Yamagata, Y. (2016).
119: (Tuel and Eltahir, 2020)
126 : (Price, R.A. 2017)
154: (Freudenberg, 2003)
156: Kotzee et Reyers, 2016 ).
164: for Batica (2015)
166: (event phase and 165 recovery phase), (Chen and Leandro (2019)
167: and recovery, (Leandro et al., (2020)
228: , 2006
231 : al., 2005
308: and H. C. Hung et al. (2016).

**References:**
425 missed :
460: not the same police
523: Nejjari A. , Abdelkader
Line 563: remove the underline
Line 565:  Plate, E. J.:.

**Acknowledgments**

Special thanks are to Dr Mohammed BEN-DAOUED, and Dr Mounir Ouzir and Mr Margaa Khalid for their meaningful insights provided

---

## Author Response (AR2)

**Point by point report**

**Reviewer 1**

We would like to extend our sincere appreciation for all of the work and dedication provided by the reviewer to improve the quality of the present work.

The reviewer requested several linguistic modifications that have been correctly addressed.

Find it very helpful. Thank you!

**Reviewer 2**

We sincerely thank the reviewer for the constructive and positive feedback. We took into consideration all the comments related to the references. As for the questions, please to find the replies in the report.

- **Questions/ answers in methodology:**

**Q1**: Households Density (HD), Old Buildings Rate (OBR) and Natural (ND) are new indicators, right? In the text, it's difficult to identify the ones that are susceptible to be taken in this study areas from literature and the new once that you considered.

**R**. We understand that this can create confusion; the table 1 identifies the references considered in this study for each indicator.

**Q2**: What are the advantages of adopting these particular case studies over others in this case?

**R.** we thank you for raising this question, discussed in the "Introduction section", the selected municipalities as case studies are experiencing a rapid urban development combined with massive problems of flooding, besides to the ancillary difficulties which lead to exponential growth in flood planning needs and endeavors. In addition, they have been reported by several studies as vulnerable to multiple climatic and non-climate hazards such as erosion and morphological changes, as well as negative process of fast urbanization. We believe that by showing all these characteristics, they represent a preferred target for the study of the resilience in coastal areas and could serve as a model for other southern Mediterranean sites facing the same issues. This was highlighted in Page 5-lines 135-140.

**Q3**: It is mentioned in .... historical records of 2014 are taken. Why are more recent data not included in the study? Are there any changes to situation in recent years?

**R.** In developing countries like Morocco, one of the most relevant challenges is data availability. Unfortunately, the 2014 census is the last official sources. Our proposed approach deals with this issue and used only available data. This was highlighted in line 364.

**Q4**: It is mentioned, "To avoid subjectivity an equal weighting was adopted". How you gave them equal weights while each parameter might have different importance level?

**R.** We adopted equal weighting considering the importance of each dimension and as mentioned to avoid subjectivity associated with the allocation of weights which can be specific to each geographic zone. In addition, our approach is built in such way that can be modified and adapted following the cases. This was highlighted in page 7 line 216-218

**Questions/ answers in results and discussion:**

**Q1:** It is mentioned, that "the high level of natural resilience is more prevalent in areas with high altitudes". Did you get any observation about this relevant? Because it's not always true, some areas with high altitude can be affected by landslide and slumping during floods.

**R.** We agree with the reviewer, not all areas of high altitudes are resilient. However, in our case, the study areas are low-lying coastal plains what led us to consider that "the high level of natural resilience is more prevalent in areas with high altitudes".

**Q2**: Are there a strong relationship between vulnerable area and resilience, can you say that all the areas with high vulnerability to the flood have automatically low resilience?

**R**. Our approach is a multi-dimensional assessment and depending on the type of vulnerability we are talking about (Physical, economic or social...) as well as the specific characteristics of the study area, this positive correlation could be true or not.

**Q3**: Can you globalize the conclusions of this work on the Nord Mediterranean Moroccan coast that are quite similar in different aspect, or is it just particularly adapted to those tree municipalities? If the response is yes, it will be interesting to mention that in your conclusions, if no please give the raison.

**R.** We are confident that our approach could be adapted successfully in different areas of the southern Mediterranean. The rest will depend on the specific geographic, economic and climatic of the studies areas, even if, for example, at the level of North Africa, the similarities are generally enough to allow the generalization of certain conclusions. This has been added in the conclusion

**Q4:** Your results are the combination of all the indicators that you establish for this work, in your opinion whish one represents the important factor for resilience in the Moroccan Mediterranean coast area?

**R.** The social and the economic components could be most significant in term of resilience enhancement in this case. But, there is no doubt that's resilience assessment could be done in a holistic way takin also into account natural and physical components.

**Q5**: I know that this subject is new research subject, especially in Nord African region, have you looked to projects in the same contexts in semi-arid coast region? It will be interesting to discuss your results with their conclusions.

**R.** In these contexts really few studies have been carried out especially in the southern Mediterranean where no studies are available. In Africa, the available work of Kotzee et al., (2016) has been done in South Africa by using a social-ecological index for measuring flood resilience as a composite index approach. Details are added in the revised version line:324-325

*Kotzee, I., & Reyers, B. (2016). Piloting a social-ecological index for measuring flood resilience: A composite index approach. Ecological Indicators, 60, 45-53.*

**Q6**: Please find bellow some corrections related to spelling

**R.** Thank you very much, we have taken seriously all comments in the new revised version of our manuscript.